

# A new index used to characterise the extent of Antarctic marine coastal winds in climate projections

Archie Cable[1], Thomas Caton Harrison[2], Elizabeth Kent[1], Richard Cornes[1], and Thomas J. Bracegirdle[2]

[1]National Oceanography Centre, Southampton, UK
[2]British Antarctic Survey, Cambridge, UK

**Correspondence:** Archie Cable (acable@noc.ac.uk)

**Abstract.** Antarctic marine coastal near-surface winds play a key role in Southern Ocean circulation. Using the ERA5 reanalysis dataset, this paper develops directional wind constancy as a tool for identifying key features in these winds and their relationship with the mid-latitude westerly jet. In particular, the Antarctic coastal wind boundary (ACWB), defined as the minimum offshore directional constancy boundary, is shown to be a useful way to define the marine near-coastal region where the Antarctic topography plays an important role in influencing the wind direction. We show that, while the ACWB is linked to large-scale modes of atmospheric circulation through its close association with variability in the mid-latitude westerly jet, it also highlights key regions where topographically-influenced, meridional flows are dominant. These meridional flows are not identified in current regional climate indices. Future changes in the ACWB are examined using CMIP6 projections for a high emissions scenario. This indicates that by the end of this century the ACWB is projected to shift poleward by about 60 km, less than the 130 km shift in the mid-latitude westerly jet, indicating a reduction in the extent of the circumpolar trough.

## 1 Introduction

Near-surface winds over marine regions close to the Antarctic coast have far-reaching impacts on the global climate. These winds help explain the structure and variability of the Antarctic Slope Current, which is a key control on the transport of relatively warm circumpolar deep water towards ice shelves, driving basal melt (Thompson et al., 2018). Additionally, the volume and extent of sea ice is strongly influenced by Antarctica's coastal winds. In some regions, coastal winds help open coastal polynyas, which are important for the formation of Antarctic bottom water (AABW). AABW in turn forms the lower limb of the meridional overturning circulation, which transports heat and carbon across the globe (Schmidt et al., 2023). Finally, winds are important for Antarctic coastal ecosystems and for polar infrastructure and operations (Baring-Gould et al., 2005).

Near-coastal Antarctic winds are linked to the strong circumpolar westerlies at mid-latitudes: the 'westerly jet'. The region between the coastal easterlies and the westerly jet is often referred to as the circumpolar trough and exhibits strong wind variability due in part to cyclones spiraling poleward. Poleward of the circumpolar trough the winds are on average dominated by easterlies. However, the easterlies are not dominant at all longitudes, with strong meridional flow at some locations (Fig. 1). This is due to the Antarctic topography, which plays an important role in controlling the coastal winds. The arrows in Fig. 1





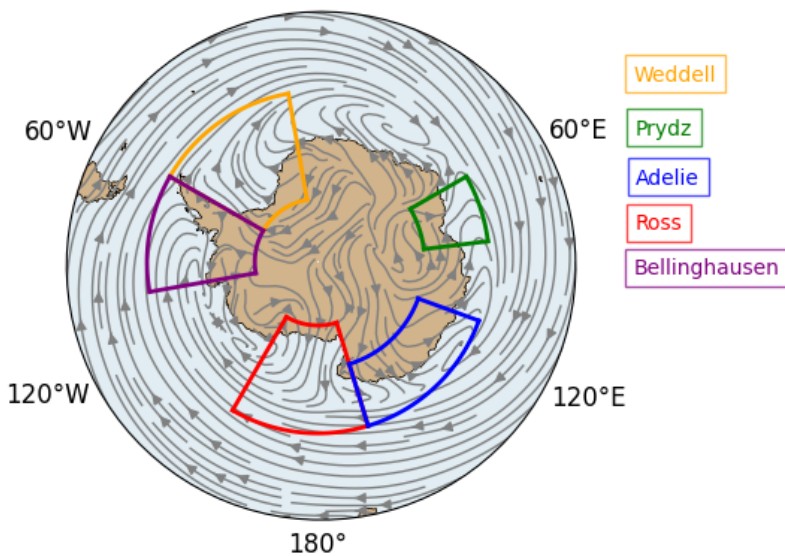

**Figure 1.** Winds around Antarctica. Grey arrows show the time-averaged winds from 1980–2023 using ERA5 monthly wind data. Highlighted regions are where extensive meridional winds prevail: Prydz Bay (60E–83E), Adélie Land (110E–163E), the Ross Sea (163E–150W), the Bellinghausen Sea (60W–100W) and the Weddell Sea (10W–60W).

represent the time-averaged wind fields from 1980–2023 using ERA5 wind data, illustrating their complex structure.

Highlighted in Fig. 1 are 5 key regions of extensive meridional winds, where the Antarctic topography is a leading driver. We refer to winds in these regions as "topographically-influenced", which encompasses a great variety of drivers (Turner et al., 2017; Goyal et al., 2021a; Orr et al., 2008; Lachlan-Cope et al., 2001). A primary example is katabatic winds, which are driven

by cold dense air in the surface boundary layer flowing down steep slopes. They prevail around much of the Antarctic coastline but are particularly extensive off Adélie Land (Parish et al., 1993; Davrinche et al., 2024) and Prydz Bay (Yu et al., 2024). Over the Ross Sea, the Transantarctic Mountains acts as a barrier to the zonal flow, creating local pressure gradients that drive the Ross Air Stream (RAS, Parish and Bromwich, 2002; Nigro and Cassano, 2014). Similarly, the Antarctic Peninsula has a strong influence on winds over the Weddell and Bellinghausen Seas (Parish, 1983; de Brito Neto et al., 2022; van Wessem et al., 2015).


Poleward shifting and strengthening of the westerly jet is a robust feature of projections using medium-high 21st century climate forcing scenarios (Goyal et al., 2021c). Although it is broadly found that coastal easterlies weaken as westerlies shift poleward (Bracegirdle et al., 2008; Neme et al., 2022), to date it has been a challenge to identify a simple index to characterise and quantify such changes. This guides the paper, which is structured around two related questions:


1. How is the extent of the Antarctic marine coastal winds projected to change in the future?



2. Can we robustly characterise the extent of these coastal winds across the entire Antarctic region?

The answer to the first question lies in the second. In order to study future projections of coastal winds and compare them to their mid-latitude counterparts, it is necessary to develop a robust way of characterising them on a circumpolar scale. In this paper we develop an approach based on directional wind constancy: a measure of how much the direction of the winds varies over a chosen period. We focus on the spatial pattern of monthly directional constancy, showing how features of the large- and local-scale winds are captured by this variable. We introduce the Antarctic coastal wind boundary (ACWB) as an index defining the latitude bounding the marine coastal winds region. We compare future projections of the ACWB and the latitude of the westerly jet to assess how the extent of the coastal winds region is projected to change in relation to the poleward-shifting westerlies.

The structure of the paper is as follows. In Sec. 2 and 3, we outline the data and methods used in our evaluation, before presenting some key features of directional constancy based on ERA5 in Sec. 4. In Sec. 4.1 and 4.2, we consider the spatial pattern of directional constancy around the Antarctic coastline and how it relates to variability in the mid-latitude winds, before introducing the ACWB and its applicatons in Sec. 4.3 and 4.4. Finally, in Sec. 5, we consider how the ACWB is projected to change by the end of the century in relation to the mid-latitude westerly jet, using CMIP6 data.

## 2 Data

### 2.1 ERA5

We use wind data from the European Centre for Medium-Range Weather Forecasts' (ECMWF) 5th generation of reanalysis datasets, ERA5 (Hersbach et al., 2020, 2023). ERA5 is based on the ECMWF Integrated Forecasting System (IFS) using 4D-Var data assimilation. The surface wind output is an instantaneous value at a height of 10 m and is diagnostically derived using boundary layer theory from winds at the lowest model level (about 40 m height). Wind data is available hourly on a $0.25° \times 0.25°$ grid from 1940 to present; however, for this work, we will use monthly-averaged data from 1980 to 2023. This gives us 43 complete years of data in the satellite era, when ERA5 will be well constrained by observations. We use ERA5 as the primary dataset for this analysis as it exhibits the best performance when reproducing scatterometer data in the Antarctic coastal region when compared with other reanalysis datasets (Caton Harrison et al., 2022). A comparison with other reanalyses is given in Appendix A.





## 2.2 Climate models

We use a 27-member ensemble of atmosphere–ocean general circulation models that participated in the World Climate Research Programme's Coupled Model Intercomparison Project Phase 6 (CMIP6; Table S1). Models are included in the ensemble if the variables needed for our analysis are available (scalar wind speed and wind components at 10 m height) as well as a suitable land-sea mask. A list of chosen models is supplied in Table S1 of the Supplementary Material. For each model we analyse the high-emissions pathway SSP5-8.5 (Eyring et al., 2016). Future changes are evaluated for the periods 2020-2039, 2040-2059, 2060-2079 and 2080-2099 for each chosen model. Where multiple ensemble members are available we select only the first one. To enable inter-model comparison, every field is interpolated onto a common 1° × 1° latitude–longitude grid spanning 90S–20S and 0–360E. Bilinear weights are computed with periodic boundaries in longitude, and models supplied on irregular native grids are mapped to a regular grid prior to interpolation. Land points are removed with each model's own land-sea mask.

## 3 Methods

### 3.1 Directional wind constancy calculation

Directional wind constancy (Singer, 1967) describes how much the wind direction varies over a chosen time period; here, we use monthly ERA5 wind data (Hersbach et al., 2020). It is defined as (Kodama et al., 1989)

$$c = \frac{\sqrt{u^2 + v^2}}{ws},\tag{1}$$

where $u$ and $v$ are monthly-mean zonal and meridional wind components respectively, and $ws$ is the monthly-mean wind speed. When $c$ is smaller than one, the wind direction varies within the month; when $c = 1$, the winds are constant in direction. This parameter therefore uses monthly data to inform us of the directional variability of the winds at higher frequencies. Whilst the directional constancy doesn't provide details of the variability, such as the temporal autocorrelation time scales, or even the direction itself, it provides information that can be used to distinguish prevailing wind regimes, and even physical drivers in some cases.

### 3.2 Other circulation and wind indices

#### 3.2.1 The Southern Annular Mode index

Large-scale atmospheric circulation in the Southern Hemisphere is broadly zonally symmetric, and is often characterised with the Southern Annular Mode (SAM, Rogers and van Loon, 1982). The SAM plays an important role in the climate of the Southern Hemisphere as it provides a combined measure of the strength and latitude of the large-scale westerly jet in the mid-latitudes. For this paper, we use the SAM index as provided by the National Oceanic and Atmospheric Administration



(NOAA), which uses 500 hPa geopotential height data from their Twentieth Century Reanalysis V2c Project (Compo et al., 2011). The SAM index is defined as the amplitude of the leading mode of variability in seasonal anomalies of 500 hPa geopotential height in the Southern Hemisphere (Mo, 2000).

### 3.2.2 Minimum Zonal Wind Boundary

The dominant zonal flow close to the Antarctic coastline is easterly; however, towards the mid-latitudes, this switches to westerly. The minimum zonal wind boundary (MZWB) denotes where this switch occurs. It is defined as the latitude of minimum over-ocean (i.e. excluding land data but including sea-ice) magnitude of zonal wind between 60S and 90S and computed here using ERA5's 10 m wind data. In the literature, it is used to define the boundary of a study region, within which Antarctic coastal winds reside (Neme et al., 2022; Caton Harrison et al., 2022; Hazel and Stewart, 2019). Note that in regions such as the Ross and Weddell Seas, zonal winds are weak and the literature uses a coastal buffer (e.g. 1000 m isobath, grid-box buffer) in these regions to account for this when defining the Antarctic coastal winds region using the MZWB.

### 3.2.3 The Jet Latitude Index

While the MZWB defines the transition zone where the time-mean zonal winds are weakest, the Jet Latitude Index (JLI) indicates the latitude at which the time-mean westerly winds are strongest. It is defined as the latitude of maximum time-mean zonal wind component between 10S and 75S at 850 hPa ($\sim$1500 m), following the definition by Bracegirdle et al. (2018). As they do, we compute it using ERA5 wind data.

### 3.3 Estimating large-scale winds

To quantify how spatial patterns of directional constancy are influenced by both local and large-scale factors, we compare directional constancy as calculated in Eq. 1 with a 'large-scale directional constancy'. Large-scale near-surface winds are derived by van den Broeke et al. (2002) and calculated from ERA5 reanalysis output in Caton Harrison et al. (2024). The large-scale term is defined as the linear extrapolation of the geostrophic wind profile at 300 hPa to 10 m under an idealised 'background' thermal wind balance. This provides an estimate of the role that large-scale pressure gradients alone play in the wind field, independent of surface-driven effects such as katabatic acceleration. The large-scale directional constancy is then calculated from these large-scale winds using Eq. 1. For more details, see Appendix B.





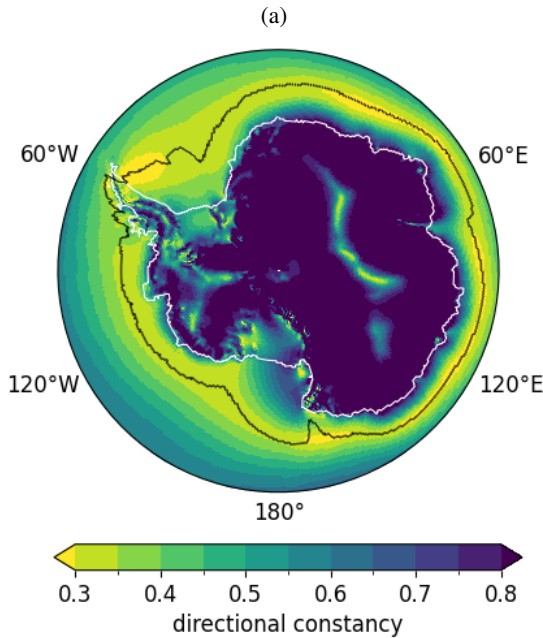
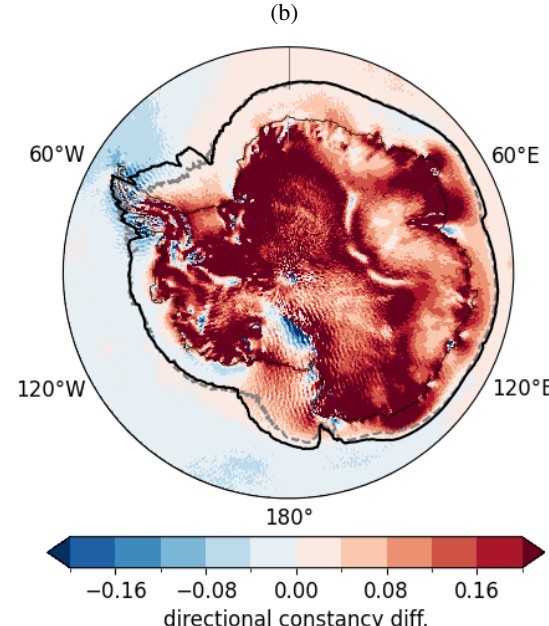

**Figure 2.** (a) The time-averaged directional constancy across the Antarctic region (60S–90S) from 1980–2023 using ERA5 wind data. Black contour indicates the time-averaged ACWB. (b) Also using ERA5 wind data, the actual minus large-scale directional constancy, time-averaged from 2010–2020 (Caton Harrison et al., 2024). Black contour shows the time-averaged ACWB; grey, dashed line shows the ACWB for the calculated large-scale wind field.

## 4 Structure of directional constancy

### 4.1 Spatial pattern of directional constancy

The 1980–2023 time-averaged directional wind constancy (Fig. 2a) shows high values along most of the Antarctic coastline, reducing to a band of low values further equatorward. We can compare this pattern with the large-scale directional constancy defined in Sec. 3.3. Fig. 2b shows the spatial differences between the actual and large-scale directional constancy, both averaged over time from 2010–2020 using ERA5 wind data. These patterns can tell us much about the different drivers of Antarctic coastal winds, leading us to establish the following broad structure:

1. Highly directionally constant winds are found over the Antarctic landmass. Here, a large radiative deficit occurs at near-surface levels, favouring high static stability and the establishment of katabatic flow over sloping terrain (Parish, 1988; Sanz Rodrigo et al., 2013; Bintanja et al., 2014; Vignon et al., 2019). Fig. 2b shows differences between the actual and large-scale directional constancy onshore, meaning a substantial portion of the actual directional constancy stems from the near-surface contribution. However, directional constancy remains high year-round despite weaker radiative cooling





in summer, suggesting that large-scale adjustment of the flow to the topography of Antarctica still plays a key role in its directional constancy (Parish and Cassano, 2003). This finding is supported by the analysis of Davrinche et al. (2024) who note that large-scale flow over Adélie Land is often aligned with katabatic flow.

2. High directional constancy also extends offshore and over ice shelves. Particularly high values ($c \sim 0.8$) occur in just a few regions, including around the western Ross Ice Shelf, Adélie Land and around western Prydz Bay. These are key sites of surface water mass transformation (Schmidt et al., 2023) and host frequent coastal polynyas during the winter months (Lin et al., 2024). Figure 2b shows that directional constancy in many offshore regions is higher than it would be under large-scale forcing alone. In the momentum budget analysis of Caton Harrison et al. (2024), these offshore regions of high directional constancy are where intense baroclinicity occurs due to diabatic cooling at low levels, which can occur even in the absence of sloping terrain (for example due to sea-ice, land-sea breezes and barrier effects). Some studies classify strong ice shelf winds as extensions of katabatic flow (Bromwich, 1989; Bromwich et al., 1993; King, 1993) although the slope of the terrain in these regions is near zero.

3. Further equatorward from the coast, directional constancy declines smoothly to a band of low values ($c \sim 0.3$). As will be discussed in more detail in Sec. 4.3, the location of the minimum directional constancy in Fig. 2a often aligns with the climatological minimum in the near-surface zonal winds: the MZWB as defined in Sec. 3.2.2. The exceptions correspond to regions of significant meridional flow. The directional constancy increases further equatorward into the region dominated by westerly flow.

This broad structure leads us to define the *Antarctic coastal wind boundary* (ACWB) as the latitude where the offshore directional constancy is at its lowest value in the region 60S–90S:

$$\text{ACWB}(t,\phi) = \theta(t,\phi)|_{\frac{\partial c}{\partial \theta}=0} \tag{2}$$

where $\theta$, $\phi$ and $t$ are latitude, longitude and time respectively. $\frac{\partial c}{\partial \theta} = 0$ indicates that we are taking the minimum value of the directional constancy as our latitude value at each time and longitude value.

The black and grey lines in Fig. 2b show the ACWB computed from the actual and large-scale directional constancy respectively. One can see that around most of the coastline, the locations are similar, meaning the large-scale behaviour dictates the position of this boundary. However, in the regions highlighted in Fig. 1, the differences become larger due to extensive meridional winds, showing where the ACWB is strongly influenced by local effects.

## 4.2 Impacts of mid-latitude variability on the spatial pattern of directional constancy

The spatial pattern of directional constancy has little variability on a seasonal basis. Some strengthening of the near-coast high directional constancy, particularly over the Ross Sea, and the low directional constancy band further offshore does occur in





austral winter compared with summer, but the effects are small and there are no prominent deviations from the band structure
identified in the time-averaged plot (Fig. 2a). Seasonal plots are provided in Fig. S1 of the Supplementary Material.


To investigate whether large-scale synoptic modes of variability affect the spatial pattern of directional constancy, we perform
a spectral decomposition using Empirical Orthogonal Functions (EOFs) on anomalies of the monthly directional constancy.
We focus on the leading mode, which captures the large-scale structure of the variability in the directional constancy anomaly.
Note that the leading mode only accounts for 25% of the variability across this large region, demonstrating the importance of
the small-scale structure of these winds. We find that there is a high correlation of up to 0.71 between the principal component
of the leading mode and the SAM index, suggesting that the leading variability in the directional constancy of Antarctic coastal
winds is closely related to the the mid-latitude westerly jet. Note that this relationship is significantly lower in higher-order
modes; for the second principal component, the correlation drops to -0.22, and drops further for higher modes of variability.


Motivated by a strong correlation with the SAM, we perform a composite analysis, focusing on East Antarctica (0–160E).
Consider the directional constancy composites based on the 10% most positive and negative leading principal component
values, shown in Fig. 3. One can see that both Fig. 3a and 3b share the same structure as described above: high directional
constancy over land (dark blue), reducing to a band of low directional constancy (bright yellow) offshore. The main difference
between the figures is the strength and position of the band of lowest directional constancy. When the leading principal com-
ponent is strongly positive (Fig. 3a), the band is close to and continuous around the coastline, although reducing in strength a
little over Prydz Bay (60E–80E) and close to the Ross Sea (∼150E). One can see this in the wind vectors (grey arrows). The
highly directionally constant winds flow northward over the ice shelf and extend a little offshore. They begin to turn toward
the west, but quickly vanish at the low band. At higher latitudes, they strengthen once again, now with a dominant westerly
component. In contrast, the strongly negative principal component (Fig. 3b) shows the low band as weaker, less continuous
and further offshore. While the onshore and katabatic winds don't change much, the south-easterly wind vectors extend further
offshore before meeting the low directional constancy band and turning into westerlies. This difference is seen quite starkly in
the wind rose diagrams (Fig. 3c and 3d), which show that the predominant offshore wind direction switches from westerlies
for high principal component values to south-east-easterlies for low values. Thus, we see that the leading mode of variability
describes the position and strength of the large-scale zonal flow in relation to the Antarctic coastline.


## 4.3 Antarctic coastal wind boundary

The ACWB, defined in Eq. (2), is a simple index that can be extracted from the directional constancy data and captures many of
the key features in Antarctic coastal winds. To understand the extent to which the different wind components affect the ACWB,
we consider their correlation over time, as shown in Fig. 4. Over the ocean, we see in Fig. 4a that the anti-correlation between
the zonal wind and ACWB is strong (<-0.5) around most of the Antarctic coastline. This is particularly strong on the eastern
coastline, where the zonal flow is largely unimpeded by the topography. This tells us that, when the westerlies are stronger,





**Figure 3.** Directional constancy averaged over dates in the top (a) and bottom (b) 10% values in the leading principal component timeseries, using monthly ERA5 wind data in the East Antarctic region (red, boxed region on right map: 0-160E, 60S–90S). Grey arrows indicate the monthly winds averaged over the same dates. (c) & (d) are the wind rose plots, based on 6-hourly wind data over the ocean, associated with (a) and (b) respectively.

the ACWB moves polewards, shrinking the region enclosed. In contrast, the correlation between the ACWB and meridional component is generally low; however, there are key hotspots, as shown by the red patches in Fig. 4b, where the correlation becomes higher (>0.5). This indicates that, when the meridional winds in these regions are strong, the ACWB moves further





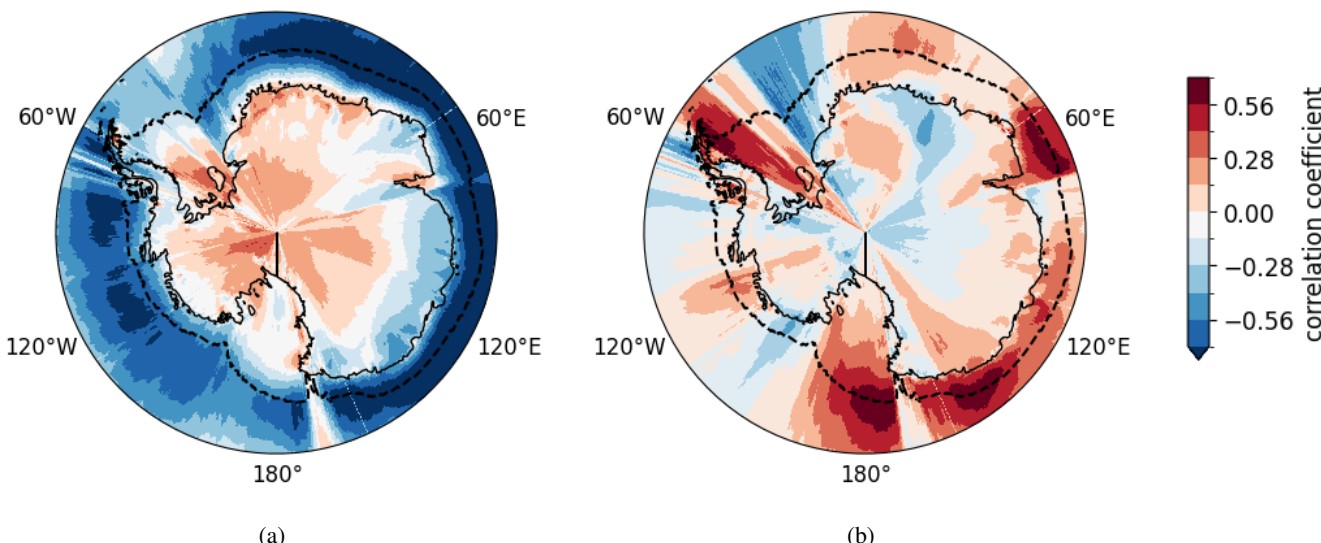

**Figure 4.** The correlation over time between the ACWB and the (a) zonal and (b) meridional component of 10 m ERA5 wind data from 1980–2023. Black dashed line indicates the time-averaged ACWB, again from 1980–2023. Jagged pattern occurs due to correlation between the ACWB and winds at a given longitude slice.

offshore. Note that these are the same regions where the large-scale directional constancy has less impact (Fig. 2b).

We can understand how the ACWB is related to the zonal wind component by comparing it with the MZWB, which defines the boundary between zonal easterlies and westerlies, as in Fig. 5. One can see by comparing the blue and red lines in Fig. 5a that the two boundaries coincide around much of the Antarctic region. This is made clearer in Fig. 5b, where the fractional 210 difference between the boundaries (red, upper line) is close to 0 at most longitudes. However, there are certain regions where the ACWB sits equatorwards of the MZWB; namely over the Weddell Sea, Ross Sea and small features offshore from Adélie Land and Prydz Bay. These are the same regions where the ACWB's correlation with the meridional component peaks (Fig. 4b) which is not captured by the MZWB, and the large-scale directional constancy has a lower impact (Fig. 2b) due to local topographic effects that extend off the coast. Here, the time-averaged zonal winds tend to be weak and thus the MZWB is less 215 spatial coherent than the ACWB. Therefore, the ACWB provides a more comprehensive definition for the Antarctic coastal wind region than the MZWB, which is commonly used in the literature, as it not only describes the location where the zonal flow moves from easterly close to the coast to westerly towards the mid-latitudes, but also highlights key locations where meridional, topographically-influenced flow off the slopes becomes significant. This avoids the problem of having to define a buffer in these regions, unlike the study region defined by the MZWB.




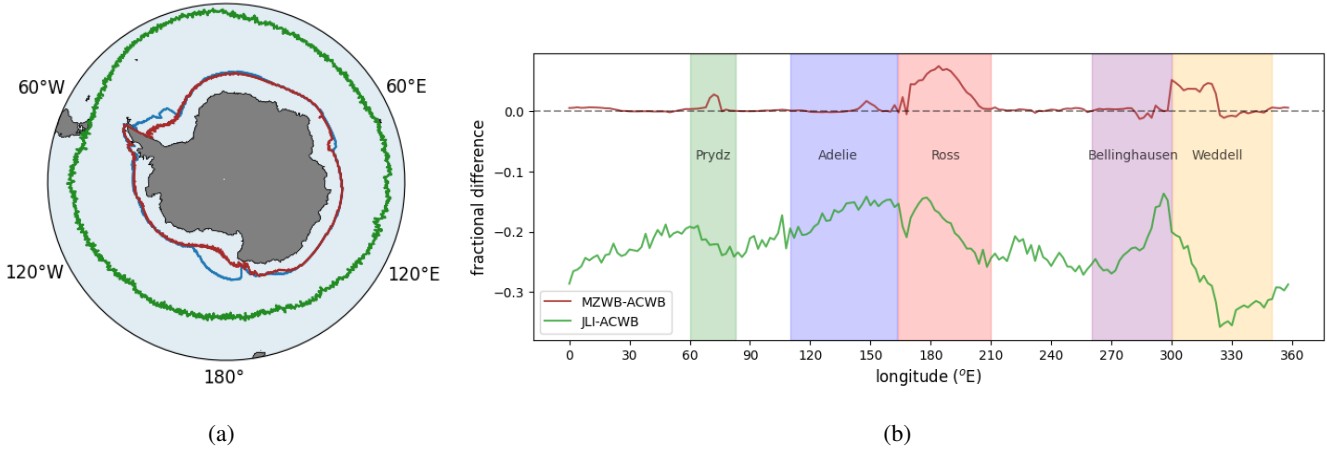

(a)                                                     (b)

**Figure 5.** (a) A contour map of Antarctica with the time-averaged ACWB (blue), MZWB (red, inner) and JLI (green, outer) from ERA5 10 m wind data and (b) the difference between the MZWB and ACWB (red, upper) and between the JLI and ACWB (green, lower), smoothed by 2° over longitude. The boxes contain the correlation over longitude between the time-averaged MZWB and ACWB (red, upper) and JLI and ACWB (green, lower). Grey dashed line in (b) is zero fractional difference.

We also compare the ACWB and the JLI and find that the fractional difference (lower green line in Fig. 5b) fluctuates with longitude. Additionally, the correlation over longitude between the ACWB and the JLI is 0.51: a non-negligible value. This highlights the relationship between the position of the mid-latitude westerly jet and the extent of Antarctic coastal winds. Note that the correlation between the MZWB and the JLI is slightly higher at 0.60, which is unsurprising as they are both defined 225 through zonal winds.

Finally, we will briefly mention the temporal correlation between the ACWB, SAM and JLI, using monthly wind data. Note that we have also compared the ACWB with other Scientific Committee on Antarctic Research (SCAR) Antarctic climate indicators, namely the Jet Speed Index (Bracegirdle et al., 2018) and the Zonal Wave Number 3 (Raphael, 2004; Goyal 230 et al., 2021b, 2022), but we find there to be very little correlation with the ACWB. However, the SAM and JLI indices have a reasonable degree of correlation with the ACWB - both 0.65 - which is similar to the correlation between the JLI and SAM (0.64). This again demonstrates the influence of the mid-latitude westerlies on the ACWB. We also consider the ACWB in two sectors: one covering the east (0–160E), the other the west (160E–0) Antarctic region. This definition includes the entire Ross Sea region in the western side. We have seen already that the east coast is very closely related to the MZWB, whereas the west 235 coast contains more topographical features, and a similar pattern manifests itself here. The correlation between the SAM and JLI indices and the eastern ACWB is higher - 0.69 and 0.67 respectively - but drops (0.51 and 0.55 respectively) when only the western coast is considered. This is because the western slopes contain two major geographic features - the Transantarctic Mountains and the Antarctic Peninsula - that interfere with the zonal flow. This again demonstrates that, while the mid-latitude





winds have an impact on their coastal counterparts, topographical influences cannot be neglected.


## 4.4 Example: The Ross Sea

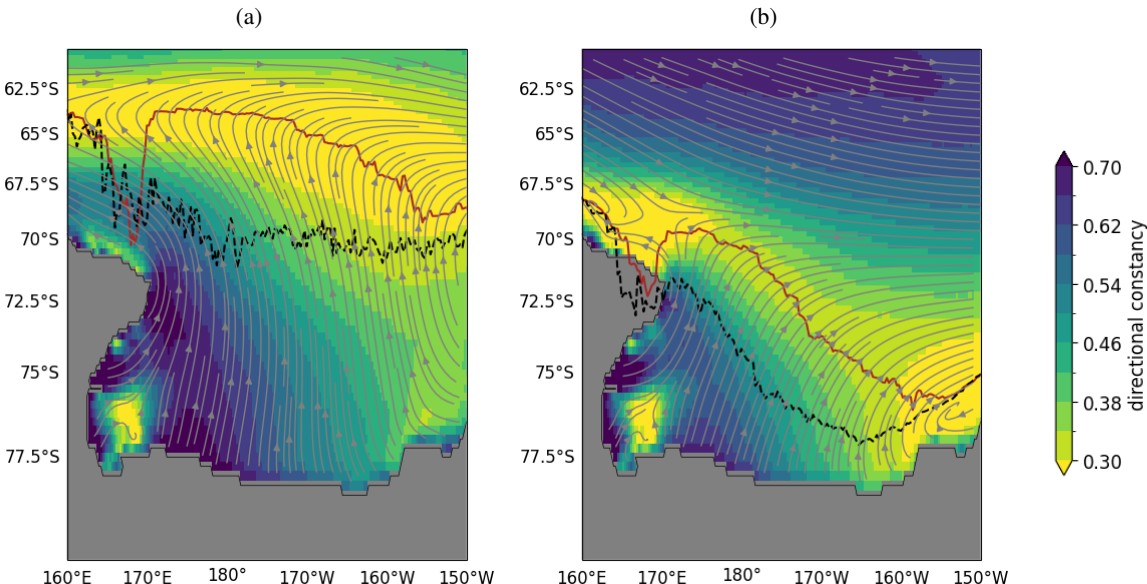

**Figure 6.** Directional constancy heat maps over dates where the area enclosed by the ACWB is, on average, in the (a) 90th and (b) 10th percentile in the Ross Sea, using ERA5 wind data. Red solid and black dashed lines are the ACWB and MZWB respectively, averaged over the relevant dates. Grey arrows are monthly wind vectors, similarly averaged over the same dates.

We have shown that the large-scale structure of the ACWB is closely related to the zonal flow around the continent except in some key regions. One such region is the Ross Sea, where the offshore winds are strongly influenced by local topography. The Transantarctic Mountains on the west provide a barrier to the easterly flow, and so winds flowing off the Ross Ice Shelf persist
further offshore on the western side of the Ross Sea, creating the Ross Ice Shelf air stream (RAS, Parish and Bromwich, 2002), before turning into westerlies around the ACWB. On the eastern coast exists a semi-permanent cyclone to the north west of Marie Byrd Land (north-eastern slopes), around 150W-160W. This Ross Cyclone is an important factor in driving the RAS (Parish et al., 2006).

We can identify these features in the directional constancy. Fig. 6a and 6b show directional constancy heatmaps, composited by the area enclosed by the ACWB in the Ross Sea in the 90th (largest area) and 10th (smallest area) percentile respectively. The main feature is the blue band of high directional constancy near the western slopes, which indicates the persistent southerly





winds of the RAS. One can see in Fig. 6a that, when the RAS is strongest it extends further into the Ross Sea and offshore. The meridional flow is maintained until it turns to the east at the ACWB, breaking through the low directional constancy contour.

Conversely, when the RAS is weakest (Fig. 6b), the blue band is confined to the western slopes and the wind speed drops to near-zero before turning to westerlies. This results in a band of low directional constancy, which is intersected by the ACWB. To the east of the RAS, the low and variable winds at the centre of the Ross Cyclone appear as an area of low directional constancy. When the meridional winds are strongest across the region, this location tends to be further offshore and more distinct (Fig. 6a), compared to when the meridional winds are weakest, where it hugs the coastline (Fig. 6b).


Note also that there is an area of low directional constancy close to the coast in Terra Nova Bay (165E,77S). This region is prone to intense katabatic wind events, rather than consistent flow, meaning the winds there are highly variable (Guest, 2021). This results in a spike of a higher latitude value in the ACWB. ERA5 picks this feature up particularly clearly due to its high resolution; other atmospheric reanalysis products don't to the same degree (see Appendix A for a comparison).


Consider now the ACWB and MZWB, which are plotted in Fig. 6 as the red and black, dashed lines respectively. In this region, they are quite different: the ACWB is much smoother and sits around the position where the dominant direction changes from southerly to westerly. Conversely, the MZWB is noisy and cuts across the RAS (Fig. 6a). It is clear that the weak zonal component of the near-coastal winds is sensitive to noise in this region, especially when the RAS is strong, whereas the

meridional contribution to the ACWB means it retains continuity. Additionally, the MZWB does not capture important aspects of the regional wind structure, again particularly when the RAS is strong, such as the flow turning west around 67S, whereas the ACWB does. This example highlights how the ACWB is an improved parameter for identifying the northern extent of Antarctic coastal winds.

## 5   Future projections

We will finally consider how the ACWB and JLI are projected to change in the future, using CMIP6 models under a high emissions (SSP5-8.5) scenario. We have found that, although there are some differences in the ACWB between ERA5 and CMIP6, the spatial pattern is very similar (Fig. C1a). We compare this in more detail in Appendix C.

We will compare 3 twenty-year future time periods - 2040–2059, 2060–2079 and 2080–2099 - with a baseline, near-term
period of 2020–2039. Fig. 7a shows the ACWB (solid) and JLI (dashed) for the near-term, 2020–2039 period (blue) and the farthest-future, 2080–2099 period (red). In both cases, the boundaries are projected to shift poleward under climate change. However, it is apparent that the JLI is expected to shift further than the ACWB, which is seen more clearly when we consider the difference between the near-term and far-future boundaries in Fig. 7b. Here, we show all three far-future periods: 2040–2059 (green, lower), 2060–2079 (orange, middle) and 2080–2099 (red, upper). We see that in both the ACWB (top) and JLI
(bottom), there is a progressive reduction in the area bounded as we move forward in time, suggesting a forced response rather





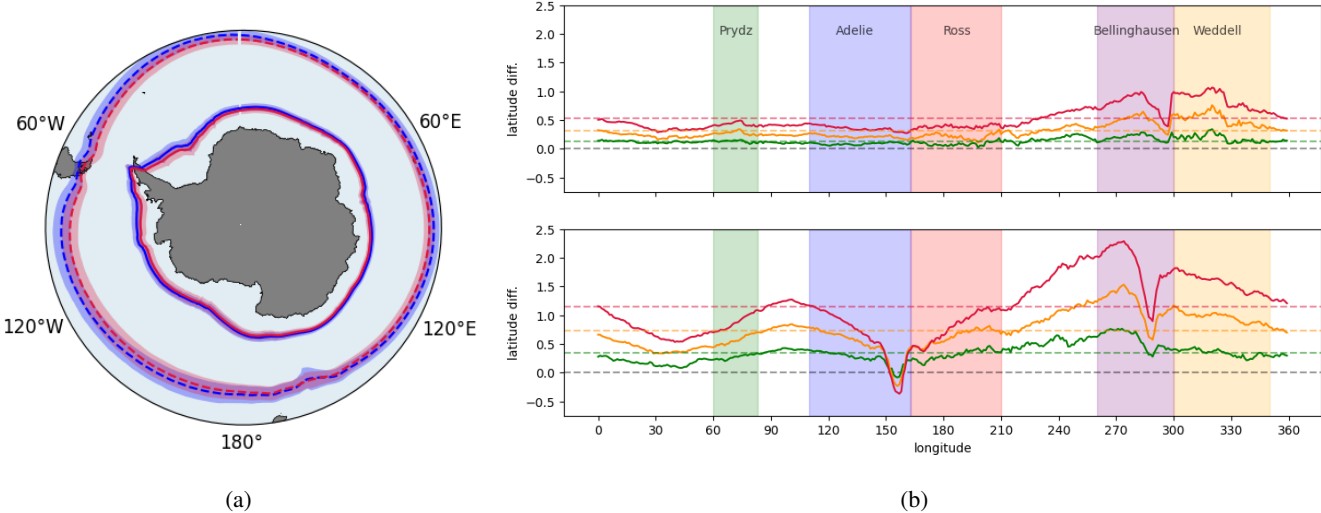

**Figure 7.** (a) The time-averaged JLI (dashed) and ACWB (solid) for the multi-model means of CMIP6, 2020–2039 (blue) and CMIP6, 2080–2099 (red). The shaded regions are the ±1 standard deviation of the CMIP6 models. (b) The difference between the 2020–2039 and 2080–2099 (red, upper), 2060–2079 (orange, middle), 2040–2059 (green, lower) models for the JLI (bottom plot) and ACWB (top plot). The dashed, coloured lines are the mean difference over latitude, while the grey dashed line is zero.

than inter-annual or inter-decadal variability. However, the ACWB shrinks far less - on average by $0.5°$ (60 km) by the end of this century - than the JLI, which on average shrinks by $1.1°$ (130km). This shows that the region of Antarctic marine coastal winds is not projected to move poleward to the same degree as the mid-latitude westerly belt. Rather, a reduction is seen in the extent of the highly-variable winds of the circumpolar trough.

Note that there is a distinction in the changes between the eastern (0–160E) and western (160E–0) areas. The eastern portion is projected to shrink less in both cases: $0.4°$ (40 km) and $0.8°$ (90 km) on the east compared to $0.6°$ (70 km) and $1.4°$ (160 km) on the west, for the ACWB and JLI respectively. It is notable that, in the western region where the Antarctic and mid-latitude winds are found to be less related, the JLI is due to shrink by almost 90 km more than the ACWB, compared to only 50 km more on the east.

## 6 Conclusions

This paper assesses how characteristics of the Antarctic coastal wind region may change in the future and develops a new index to quantify this. It is based on directional wind constancy: a parameter that describes the directional variability of winds over a chosen time period. By combining zonal and meridional wind components, it helps to distinguish between two re-



gions of broadly directionally constant flow: the mid-latitude westerly jet and the topographically-influenced Antarctic coastal winds. The latter have a strong easterly component driven by large-scale pressure gradients, but are also strongly influenced by Antarctic topography. We have shown that directional constancy is sensitive to both these drivers. The large-scale pattern is that of high directional constancy near the shore and low directional constancy further offshore, highlighting the location

where the average zonal wind direction switches from easterly in the Antarctic region to westerly in the mid-latitudes. Highly directionally-constant winds in certain marine regions - namely the Weddell and Bellinghausen Seas, the Ross Sea, Adélie Land and Prydz Bay - disrupt the coastal easterlies and are a result of the influence of topography on both large-scale and mesoscale pressure patterns.

Given this circumpolar band-like structure, we define the Antarctic coastal wind boundary (ACWB) as the latitude of minimum offshore directional constancy. In general, it defines the boundary where the mid-latitude westerlies switch to coastal easterlies, except in the identified key regions where the meridional flow dominates. Here, it correlates strongly with the meridional wind component. This better identifies the northern boundary of Antarctic coastal winds, compared to the minimum zonal wind boundary (MZWB) typically used in the literature, because it captures the full extent of these topographic flows. A clear

example of this is in the Ross Sea, where the ACWB contours the Ross Ice Shelf air stream, whereas the MZWB cuts through it.

In order to evaluate the stability of the ACWB under climate change, we considered how it is related to well-known changes to mid-latitude climate indices, such as the Jet Latitude Index (JLI) and the Southern Annular Mode (SAM). We have seen that, while the large-scale structure of coastal winds is strongly influenced by mid-latitude drivers, topographical drivers cannot be

ignored as they give rise to important local features. In particular, the ACWB is useful as a near-coastal climate index, which can be compared to equivalent mid-latitude indices like the JLI.

We put this into practice to study how the Antarctic coastal wind region is projected to change by the end of this century in comparison to the poleward-shifting westerly jet. We have computed the ACWB using CMIP6 climate models in a high

emissions scenario, and compare it with future projections of the JLI. We find that, while the westerly jet is projected to shift southward by 130 km on average, the Antarctic region is more stable, only shrinking by 60 km. This suggests that, although the westerly jet is closely connected to the northern extent of Antarctic coastal winds, other regional factors may limit their southward contraction under climate change. This further suggests a reduction in the spatial extent of the highly variable winds of the circumpolar trough region. This work provides a platform for studying the structure and strength of the winds within the

ACWB, and how they will change in the future.

Antarctic coastal winds are major drivers of Southern Ocean circulation and sea-ice variability. We have described an index that can be robustly calculated across model datasets, including future projections. This index can be used to relate the mid-latitude westerly jet directly to Antarctic coastal winds. Our analysis suggests that, although the two are closely related, they




cannot be conflated. Understanding the relationship between mid-latitude and polar winds will be important for constraining future projections of the Antarctic climate.

## Appendix A: Comparison of reanalysis datasets

For this paper, we have used the ERA5 wind data as there is evidence to suggest it does the best job at capturing Antarctic coastal winds when compared to other reanalysis products (Caton Harrison et al., 2022). However, it is still worth comparing
some of our results with 3 additional reanalysis datasets - JRA3Q (JMA, 2023), JRA55 (JMA, 2013) and MERRA2 (GMAO, 2015). Fig. A1 shows the fractional differences between the directional constancy climatology of the 4 reanalyses.

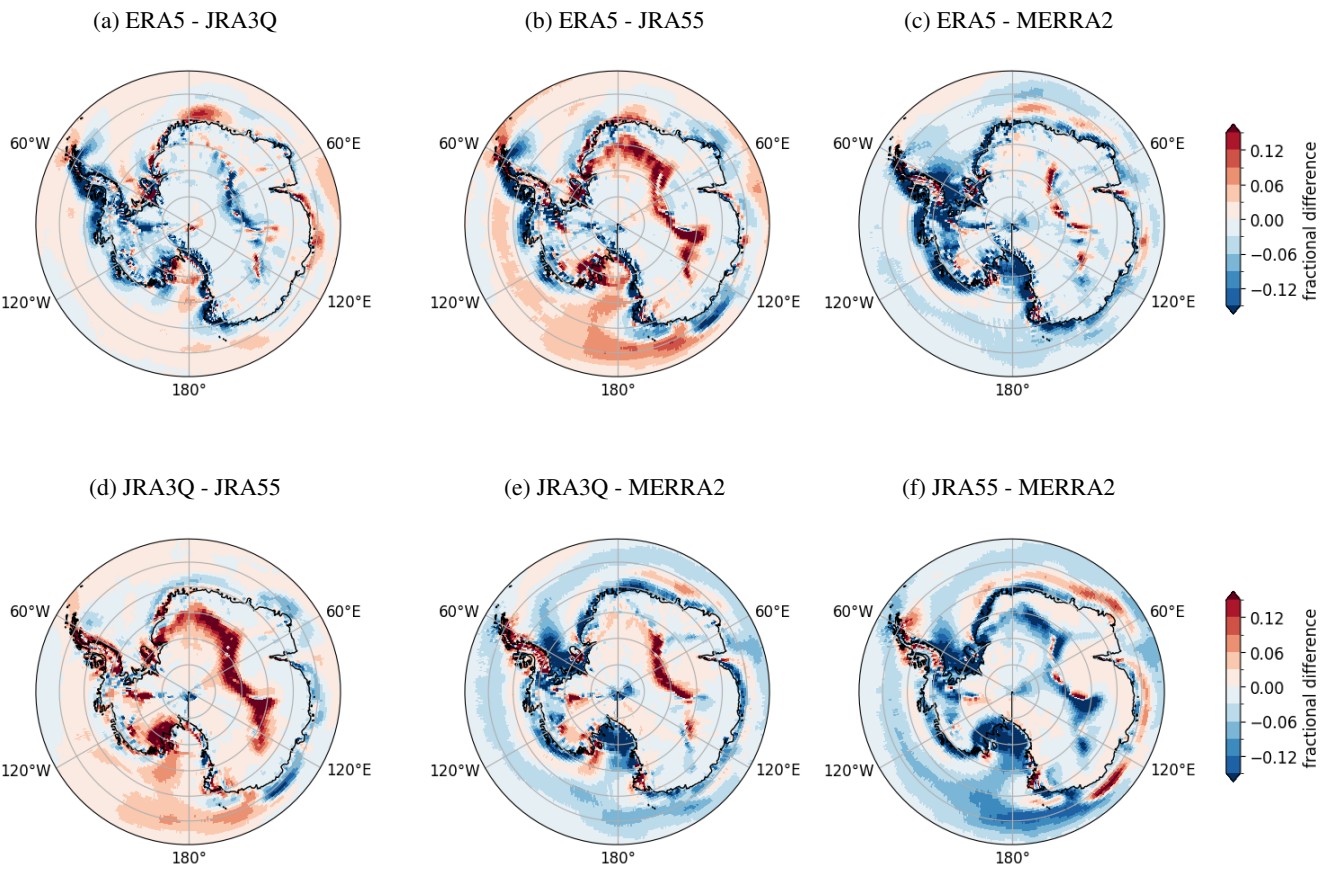

**Figure A1.** The fractional difference of the mean directional constancy, calculated from monthly wind data, between the 4 reanalysis products. Fractional difference calculated, for example, between ERA5 & JRA3Q as $\frac{2(ERA5-JRA3Q)}{ERA5+JRA3Q}$.



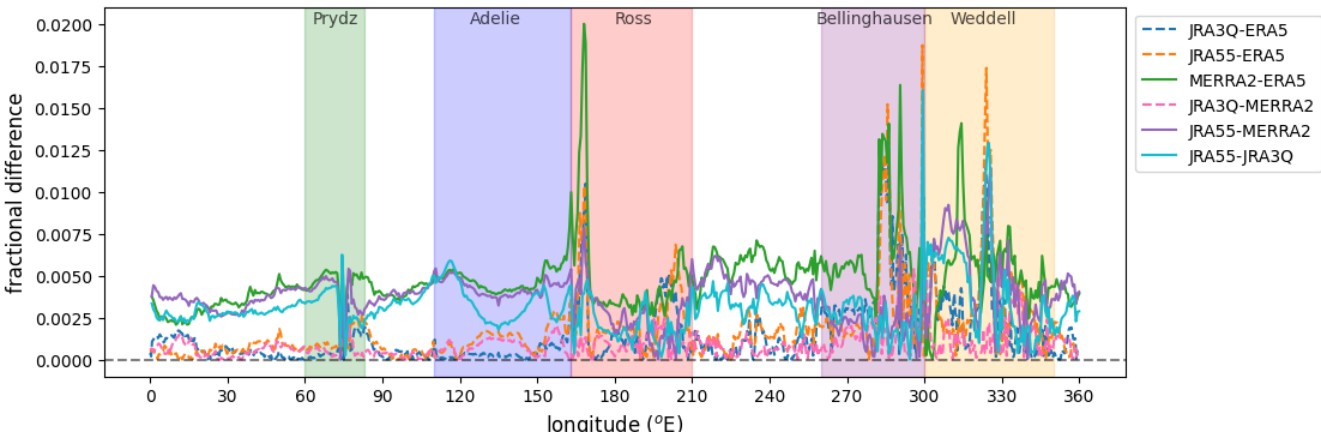

**Figure A2.** The absolute fractional difference (defined as in Fig. A1) between reanalysis datasets (coloured lines) for the time-averaged ACWB. Grey, dashed line indicates zero fractional difference.

Over the ocean, the agreement is generally good, with less than 0.05 absolute fractional difference in most locations, especially further offshore. However, near the coastline, the differences become more pronounced, particularly on the west side.
MERRA2 predicts consistently higher directional constancy along the western coastline, particularly over the Ronne (in the Weddell Sea) and Ross Ice Shelves. These differences become more pronounced when comparing with the higher resolution ERA5 data. Indeed, the higher resolution data tends to predict a lower average directional constancy along the western coastline, particularly around the Antarctic Peninsula. It is likely because the higher resolution and improved representation of the orography picks up smaller-scale events that cause higher wind variability across a localised region.


We also compare the absolute fractional difference of the time-averaged ACWB between the reanalyses, as in Fig. A2. On average, the agreement tends to be quite good, with an absolute fractional difference of up to 0.007 at most longitudes. However, it is clear that MERRA2 differs the most from the other three reanalyses, as shown by the solid lines. This matches the behaviour shown in Fig. A1. Again, it appears that the lower resolution data fails to pick up localised, near-coastal events,
which affects the ACWB.

In the Ross Sea, there is a sharp spike where the data disagree quite strongly. In particular, ERA5 predicts the ACWB to be at further south than the other reanalyses at around 165E. This is because it picks up an area of particularly low directional constancy in Terra Nova Bay (165E,77S), as seen in Fig. 6.


Finally, the ACWB across the Antarctic Peninsula (between the Weddell and Bellinghausen Seas) differs quite a lot across the reanalysis products. One can see from Fig. A1 that the high resolution products predict a lower directional constancy near





the coastline, which is affecting the ACWB here. For example, there is a spike of difference between ERA5 and JRA55 on the
west side of the Weddell Sea, where ERA5 predicts a region of low directional constancy at the tip of the Peninsula, a feature
that is not so prominent in JRA55.

Overall, one can see that the resolution of the reanalysis model contributes significantly to the location of the ACWB and
to the structure of directional constancy very near the coastline. This supports conclusions drawn by Caton Harrison et al.
(2022), who stated that ERA5 exhibits the best overall performance in regions where conditions favour katabatic forcing when
compared to scatterometry data. We conclude that the higher resolution of ERA5, compared to the other reanalyses, plays an
important role in this improved behaviour.

## Appendix B: Large-scale winds

The large-scale directional constancy is computed following Caton Harrison et al. (2024) and summarised here. First, the
geostrophic horizontal wind at 300 hPa height, $u_g$ and $v_g$, is calculated:

$$u_g = -\frac{g}{f}\frac{\partial \Phi}{\partial y}, \qquad v_g = \frac{g}{f}\frac{\partial \Phi}{\partial x}, \tag{B1}$$

where $\Phi$ is geopotential height, $g$ gravity and $f$ the Coriolis parameter. Next, vertical shear between 300 hPa and the surface
pressure $p_s$ is supplied by the thermal-wind relation using the background potential-temperature field $\theta_0$:

$$\frac{\partial u_{\mathrm{lsc}}}{\partial \ln p} = \frac{R_d}{f}\frac{\partial \theta_0}{\partial y}, \qquad \frac{\partial v_{\mathrm{lsc}}}{\partial \ln p} = -\frac{R_d}{f}\frac{\partial \theta_0}{\partial x}, \tag{B2}$$

where $u_{lsc} = u_g$ and $v_{lsc} = v_g$ at 300 hPa, and $R_d$ is the gas constant. Equation B2 is integrated to the surface and interpolated
to 10 m height to obtain $u_{10,lsc}$ and $v_{10,lsc}$ (large-scale, near-surface winds). The background potential temperature $\theta_0$ repre-
sents a smoothed profile which is equal to potential temperature except near the surface, where real potential temperature is
sharply reduced due to radiative cooling (known as the temperature deficit). Background potential temperature is obtained by
linearly extrapolating potential temperature from upper levels to the surface. Full details of the temperature-deficit formulation
and its evaluation are given in Caton Harrison et al. (2024). Finally, $u_{10,lsc}$ and $v_{10,lsc}$ are substituted into Eq. 1 to obtain
large-scale directional constancy.

## Appendix C: Comparison of ERA5 reanalysis and climate models

We also compare ERA5 with the CMIP6 historical model from 1980–2010 (as described in Sec. 2.2), to test the performance of
global climate models in capturing these boundaries and representing Antarctic marine coastal winds. We do this by comparing



the ACWB and JLI.

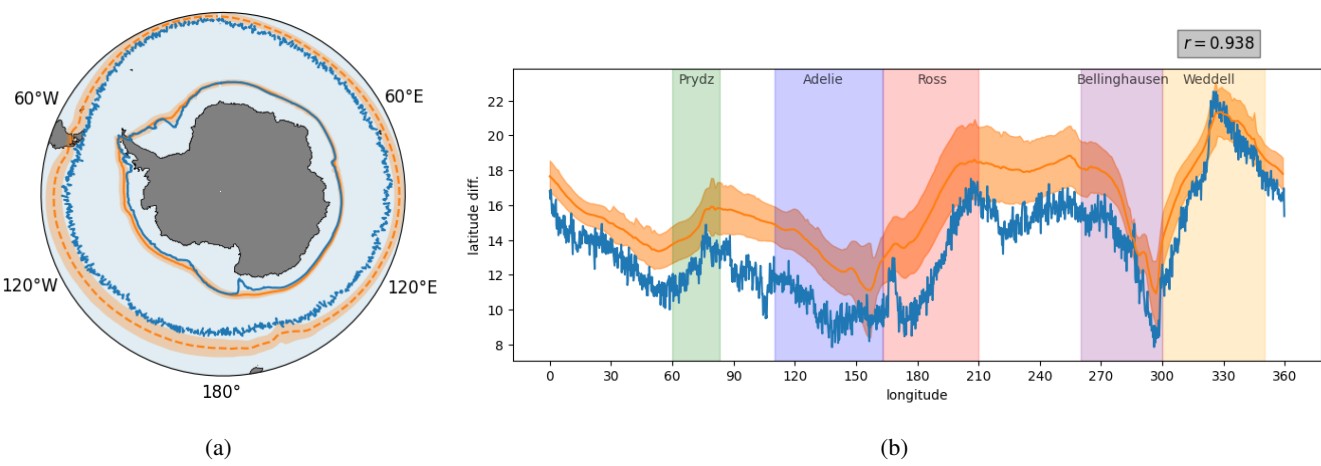

(a)                                                          (b)

**Figure C1.** (a) The time-averaged JLI (dashed) and ACWB (solid) for ERA5 (blue) and CMIP6 historical multi-model mean (orange) from 1980–2010, and (b) the difference between the JLI and ACWB in both estimates. The shaded orange region is the $\pm 1$ standard deviation of the CMIP6 models. The number in the grey box is the correlation over longitude between the two lines in (b).

Fig. C1a shows the two boundaries plotted around a map of Antarctica for ERA5 (blue) and the CMIP6 multi-model mean (orange). Consider first the JLI (dashed lines), which indicates the position of the strongest westerlies. The climate models consistently predict the JLI to be at lower latitudes than ERA5 at all longitudes. However, the correlation over longitude between the two is very high at 0.95. This suggests that the climate models are picking up the same spatial pattern in the westerlies as ERA5, but the whole position is slightly shifted away from the pole.

For the ACWB (solid lines), the climate models and ERA5 agree quite well along most of the coastline, particularly on the eastern side. Over the Amundsen and Bellinghausen Seas to the west of the Peninsula, the climate model predicts consistently lower latitudes than ERA5, with differences of order 1 standard deviation from the multi-model mean. The other exceptions are a small area around the Weddell Sea and to the west of the Ross Sea. It is probable that, in both cases, the higher resolution of ERA5 (0.25° as opposed to 1° for CMIP6) gives rise to features at the coastline that cause the ACWB to dip closer to the coast. Note also that we see a slight dip in the JLI at around 170E in CMIP6, which is caused by the split jet at that location (Chiang et al., 2018). Similar to the JLI, the correlation over longitude between the ACWB in climate models and ERA5 is extremely high at 0.94. This is encouraging as it suggests that the climate models are computing the same spatial pattern in the Antarctic coastal winds - both large-scale and topographically-influenced - as in ERA5, at least in the recreation of these boundaries.



For the projections, we will also compare the change in the JLI against those in the ACWB. Fig. C1b shows the difference between the JLI and ACWB for ERA5 (blue) and the historical CMIP6 multi-model mean (orange). One can see that the latitudinal difference betwen the ACWB and JLI is, on average, slightly larger in CMIP than ERA5. However, importantly, the correlation between these two lines is very high - 0.94 - and so, while ERA5 and CMIP6 disagree on the precise difference, they agree on the general behaviour, which is most important for our purposes. Thus, we can be confident that we will be

capturing the correct variability in these boundaries using CMIP6, even if the precise latitudes are slightly different.

*Supplement.* The supplement related to this article can be found online at https://doi.org/10.5194/wcd-0-1-2025-supplement

*Author contributions.* AC with EK developed the initial concept for the analysis with further development from TCH, RC and TB. AC

developed the ACWB index and analysed the reanalyses. TCH analysed the climate models and made the large-scale directional constancy calculations. All authors contributed to the revision of the initial paper draft by AC, and all authors have read and approved the submitted version.

*Competing interests.* The contact author has declared that none of the authors have any competing interests

*Data availability.* ERA5 was downloaded from the Climate Data Store at https://cds.climate.copernicus.eu/datasets (Hersbach et al., 2023).

MERRA-2 was downloaded from the Goddard Earth Sciences Data & Information Services Center at https://disc.gsfc.nasa.gov/datasets?project=MERRA-2 (GMAO, 2015). JRA-55 and JRA-3Q were both downloaded from the National Science Foundation (NSF) National Center for Environmental Research (NCAR) Research Data Archive at https://rda.ucar.edu/lookfordata/ (JMA, 2013, 2023). CMIP6 model data was accessed from the Earth Systems Grid Federation (ESGF) website at https://esgf-node.llnl.gov/search/cmip6/

*Acknowledgements.* AC, EK, TB and TCH were funded by NERC grant "Improved projections of winds at the crossroads between Antarctica

and the Southern Ocean" (NE/V000969/1 at NOC and NE/V000691/1 at BAS). RC recieved support from the NOC National Capability Program AtlantiS (NE/Y005589/1). We are grateful to the European Centre for Medium-range Weather Forecasts (ECMWF), the NASA Global Modeling and Assimilation Office (GMAO) and the Japan Meteorological Agency (JMA) for making ERA5, MERRA-2 and JRA-55 & JRA-3Q respectively. We acknowledge the World Climate Research Programme, which, through its Working Group on Coupled Modelling, coordinated and promoted CMIP6. We thank the climate modeling groups for producing and making available their model output, the Earth

System Grid Federation (ESGF) for archiving the data and providing access, and the multiple funding agencies who support CMIP6 and ESGF.



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
