# Peer review of "A new index used to characterise the extent of Antarctic marine coastal winds in climate projections"

_EGUsphere, 2025_

## Referee Comment (RC1)

In this paper, authors defined the Antarctic coastal wind boundary (ACWB) as the latitude of minimum offshore directional constancy to distinguish coastal wind (easterly winds) from westerly winds. The index also is suitable for the region with strong meridonal winds. The paper was written well. The novelty of the manuscript is great. The manuscript only examined the impacts of the westerly jet (SAM index) on the ACWB index. How the factors over the continent (katabatic winds) influence the ACWB index needs to be assessed. Some specific concerns should be addressed before the consideration of the accepting the manuscript.

1. Across the entire manuscript, authors compared the new index (ACWB) with the previous index (MZWB). A summary (table) is encouraged to show the difference of them.
2. Authors may consider show the variation of the new index, from seasonal to trend.
3. More emissions scenarios may be considered to show the change with greenhouse increase.
4. What percentage different factors (jet or continent warming) can explain the trend in the ACWB index in the future?
5. Maybe authors may discuss the relationship between the ACWB index and sea ice variation.

---

## Author Response (AR1)

**Referee Comments**

November 26, 2025

We thank both referees for their insightful comments. We have responded to each comment below and have added to the manuscript accordingly, which has improved the quality of the revised paper.

We provide a list of the changes made to the manuscript in response to the referee and editor comments below, followed by a detailed response to the referee comments. All line, section and figure references relate to those quoted in the preprint unless otherwise stated.

- Added a new subsection describing the ACWB to the start of Sec. 3.2.

- Moved subsection on MZWB (Sec. 3.2.2) above the subsection on the SAM (Sec. 3.2.1).

- Added Table 1 at the end of Sec. 3.2, summarising the different climate indices used in this paper.

- Line 122: added "above the surface" after 300 hPa to 10 m.

- Fig. 2: change "actual minus large-scale" to "difference between the actual and large-scale".

- Line 153: added "Beyond the MZWB" before "the directional constancy...".

- Lines 155-160: removed.

- Line 161: added "defined in Sec. 3.2.1" after "show the ACWB".

- Line 164: added sentence beginning with "Here, topographical flow extends...".

- Add new paragraph at end of Sec. 4.1 describing the temporal variability.

- Removed first paragraph of Sec. 4.2.

- Line 188: changed "higher" to "lower".

- Line 208: changed "blue" to "black".

- Line 221: changed "green" to "blue".

- Line 226: added sentence "All correlations mentioned...".

- Fig. 6: removed land mask and extended the region to see what the winds look like onshore.

- Line 252: added a few comments to this sentence about the winds over the land.

- Line 277: added sentences starting with "We consider the highest emission scenario...".

- Line 284: changed "green" to "blue".

- Line 289: added the sentence "Such a reduction...". This includes a new reference (Patterson et. al., 2019) added to the bibliography.

- Added seasonal and inter-decadal plots to the Supplementary Material.

- Added Sec. S3 in the Supplementary Material entitled "Future projections for different emissions scenarios", which includes an additional plot.

- Fig. 5, 7 and A2 have been updated to allow readers with colour vision deficiencies to correctly interpret your findings. Updated the captions accordingly.

**1 Referee 1**

1. *How the factors over the continent (katabatic winds) influence the ACWB index needs to be assessed.*

   Continent winds are highly influential on the ACWB, but predominantly in the 5 key regions highlighted: Prydz Bay, Adelie Land, Ross Sea and Bellinghausen and Weddell Seas. As the ACWB encloses the Antarctic marine coastal winds, only extensive topographical flow will have a direct impact on its location. Otherwise, the large-scale easterlies dominate this behaviour. This can be seen when comparing the actual and large-scale directional constancy and ACWB in Fig. 2b. There is very little difference between the large-scale and actual ACWBs (black and grey lines) most of the way around the continent, indicating that the large-scale drivers dominate. However, in the highlighted regions, there is a substantial difference: the local-scale drivers create an extended flow, which interrupts the large-scale easterlies.

   We have added more clarity on this in our discussion of Fig. 2b (last paragraph of Sec. 4.1).

2. *Across the entire manuscript, authors compare the new index (ACWB) with the previous index (MZWB). A summary (table) is encouraged to show the difference of them.*

   We agree this would help improve the clarity. We have added a table to Sec. 3.2, which includes a description of all the climate indices used in the paper.

3. *Authors may consider show the variation of the new index, from seasonal to trend.*

   The spatial structure of directional constancy doesn't change substantially between seasons, and the seasonal variability in the ACWB is far lower than the month-by-month variability. Note that there is very little seasonal variability at all on the east coast and over the Ross Sea; however, there is more seasonality on the west coast, with higher latitude positions in the shoulder seasons, which could be related to the spring and autumn poleward contraction and strengthening of the mid-latitude storm track in association with the semi-annual oscillation in sea-level pressure (van Loon, Rogers, 1984). Additionally, there isn't a clear trend for the period 1980-2023 in the ACWB, based on looking at interdecadal variability.

   In response to this comment, we have added a plot of the seasonal and interdecadal ACWBs to the Supplementary Material and have slightly extended our discussion of the variability in the final paragraph of Sec. 4.1 to clarify both the seasonal and multi-decadal behaviours.

4. *More emissions scenarios may be considered to show the change with greenhouse gas increase.*

   This is an interesting question so thank you for raising it. In response to this comment, we have compared 3 greenhouse gas emission scenarios - SSP2-4.5, SSP3-7.0 and SSP5-8.5 (used in the paper) - in Figure 2 below. We see that the same behaviour of southward-shifting boundaries occurs in all scenarios, but to a higher extent in high emission scenarios. This suggests that emissions are a key factor in the southward shifting of both these boundaries.

   We have added Figure 2 to the Supplementary Material and added a comment in the manuscript.

5. *What percentage different factors (jet or continent warming) can explain the trend in the ACWB index in the future?*

   This is an interesting but challenging question to resolve because it requires unpicking the many different factors that drive Antarctic winds, which goes beyond the scope of this work. We note that the EOF analysis performed in Sec. 4.2 shows that the leading mode, which accounts for 25% of the variance, is strongly correlated with the SAM index, which is itself closely linked with the mid-latitude jet. Higher-order modes then capture different behaviours, such as the local-scale impacts, but become harder to separate into specific physical drivers.

   We have not revised the manuscript in response to this comment.

6. *Maybe authors may discuss the relationship between the ACWB index and sea ice variation.*

This is an interesting comment and something we considered when preparing this manuscript. However, it is challenging to provide a direct link between sea-ice and the ACWB because there are a huge number of factors affecting its behaviour. Our focus is to define a region, whereas understanding links between directional constancy and sea-ice would require closer consideration of behaviours within the boundary. Thus, we concluded it was beyond the scope of this paper.

**2 Referee 2**

1. *Methods: While I appreciate the narrative logic of defining the Antarctic coastal wind boundary (ACWB) in Section 4, I feel like it would be clearer for the reader if it was defined in the Methods section (Section 3). Seeing it defined just after or before the minimum zonal wind boundary (MZWB) would make it easier to differentiate the two when they are discussed together later in the text.*

We agree that this would be clearer, so we have added a subsection in Methods outlining the ACWB.

2. *Section 4.4: The highlight for the Ross Sea is compelling, but it could be interesting to see the analysis extended over the Ross Ice Shelf as well. When looking at Figure 2, several interesting features appear over the ice shelf. You see the Ross Ice Shelf Airstream along the Transantarctic Mountains and the barrier wind corner jet near the Prince Olav Mountains. If you want, you can incorporate the difference in actual and large-scale directional consistency as shown in Figure 2b to show how your analysis also highlights the influence of polar jet stream/westerlies in controlling these inland Antarctic features. (Nigro et al. 2012; Seefeldt and Cassano 2012).*

We agree that this is interesting and have removed the landmask and extended the region shown in the plot to 150E so that one can see the winds flowing off the Transantarctic Mountains. This highlights the origin of the extended topographic flow. We've added a few comments in the main text to reflect this.

3. *If it is not computationally expensive, it could be interesting to see how future climate projections in the position of the MZWB are different than the ACWB. This may potentially highlight the different future climate interpretations that come from utilizing the ACWB instead of the MZWB.*

We have considered this and have attached Figure 1 for your interest (see last page), which is Fig. 7 from the paper with the MZWB added. One can see that the future projection of the MZWB is very similar to the ACWB i.e. it is expected to shift southward, but not to the same extent as the JLI. Additionally, the deviation of the MZWB from the ACWB in our key regions is consistent across the different future time periods. Thus, the key difference between the two remains in the topographical regions and this is not projected to change in the future. We have chosen not to include this in the paper for clarity; the key is to highlight the different projections of the ACWB and JLI.

Minor comments from Referee 2:

- *Line 10: If there is space, perhaps include a broader implication about what a reduction in the extent of the circumpolar trough means for the Antarctic climate*
  Our response has been combined with that for the comment on Line 328-329 below.

- *Line 122: Do you mean 10 m above the surface? Please clarify.*
  Yes, added a clarification in the paper.

- *Fig. 2: For clarity, I suggest replacing "the actual minus large-scale directional constancy" with "the difference between the actual and large-scale directional constancy"*
  Thanks, we've changed the text accordingly.

- *Line 153: Add, "Beyond the MZWB, ..." before the sentence "The directional constancy increases further equatorward into the region dominated by westerly flow."*
  Added.

- *Fig. 3a and 3b: Is it possible to slightly extend the northern bound of the figures to show more of the ocean and thus the westerly jet?*
  Bounds extended.

- *Line 188: Assuming you are referring to the westerly winds north of the low consistency band, then you should say at "lower latitudes" than "higher latitudes". Higher latitudes refer to getting closer to the Poles.*
  Fixed.

- *Line 227-239: Please mention if these correlations are statistically significant.* They are (checked with p-value).
  Added a comment in the paper.

- *Figure 7: Please include a JLI and ACWB label for the two boxes.*
  Labels added

- *Line 328-329: Could you possibly infer/provide an educated guess as to what could be the effects on the Antarctic climate from this projected mismatch in poleward contradiction between the westerly jet and the ACWB?*

  We will respond to the *Line 10* comment (*what a reduction in the extent of the circumpolar trough means for the Antarctic climate*)and this one simultaneously as they are two sides of the same coin. The effects of the future projections, both to winds in the circumpolar trough and the Antarctic climate, are very interesting and we thank you for raising this.

  The simple answer is that these behaviours are highly dependent on processes within these regions - the circumpolar trough and the Antarctic regions - and simply studying the boundary between them isn't sufficient to understand them.

  The literature describes two potential consequences of a reduced circumpolar trough region. Russell et. al (2018) examined westerly winds in multiple model simulations, and they found that models with a narrower westerly jet (which implies a narrower circumpolar trough) showed increased net heat and carbon uptake. Additionally, studies by Patterson et. al. (2019) project a reduction in atmospheric blocking in the mid-latitudes, which could see a reduction in extreme weather events.

  In response to this comment, we have added a sentence highlighting these two papers at the end of Sec. 5

[Figure]

(a)

(b)

Figure 1: (a) The time-averaged JLI (dashed), MZWB (dotted) and ACWB (solid) for the multi-model means of CMIP6, 2020–2039 (blue) and CMIP6, 2080–2099 (red). The shaded regions are the ±1 standard deviation of the CMIP6 models. (b) The difference between the 2020–2039 and 2080–2099 (red, upper), 2060–2079 (orange, middle), 2040–2059 (blue, lower) models for the JLI (bottom plot) and ACWB (top plot). The dashed, coloured lines are the mean difference over latitude, while the grey dashed line is zero.

[Figure]

Figure 2: A comparison of future projections based on different greenhouse gas emission scenarios for the ACWB (upper plot) and JLI (lower plot). Plots show the latitudinal difference between the periods 2020-2039 and 2080-2099 for each boundary. Blue, orange and red lines show the SSP2-4.5, SSP3-7.0 and SSP5-8.5 scenarios respectively. Coloured, dashed lines show the longitudinal mean of this difference, while the grey, dashed line runs through 0 difference.

---

## Author Response (AR3)

**Response to Editor comments**

January 5, 2026

In response to the following Editor comment:

*Thank you for submitting your manuscript to Weather and Climate Dynamics. The reviewers were positive about the paper's clarity and scientific contribution, and I agree with their assessments.*

*I have only one minor editorial comment. The analysis focuses on all-months relationships between the ACWB, SAM, and JLI. While seasonal directional-constancy maps are included in the Supplement, the manuscript does not examine whether these large-scale relationships differ by season. Briefly exploring how the ACWB–SAM and ACWB–JLI relationships might change seasonally would strengthen the interpretation.*

*I'd be happy to accept the manuscript for publication after clarifying this point.*

We thank the Editor for their comments.

The seasonality of the ACWB and other indices is something we did analyse when preparing the manuscript but it is a complex matter given the various drivers involved. We decided that that the seasonal variations were not the most important results, in what was already a long paper. We considered it would substantially increase the length of the paper to provide additional text and figures to support a robust description of the seasonal effects. We have however added a short paragraph at the end of Sec. 4.3 to give an indication of how the seasonality of the ACWB is influenced by the seasonal variation in the local- and large-scale drivers.

Updated on 05/01/2026

In response to the following Editor comment, which follows on from the previous comments:

*Thank you for your careful response and for adding a paragraph clarifying the role of seasonality in Section 4.3. I agree that presenting a full seasonal breakdown of all analyses in the main text would substantially increase the length of the paper. However, I still believe that including a seasonal version of Figure 5 in the Supplementary Material would add significant value to the manuscript. While the seasonal variations in ACWB are modest, the relationships with the large-scale indices are seasonally dependent (as you mentioned), and a dedicated figure would help clarify this point for readers.*

We thank the Editor for the comment and have now added a seasonal version of Fig. 5 in the Supplementary Material (S4).

*In addition, please explicitly define how fractional difference is calculated in the Methods section. At present, the definition appears only in the caption of Fig. A1, which makes it difficult for readers to locate. Please also ensure that the definition is presented in a general form, rather than being limited to the specific reanalysis comparison shown in that figure.*

We thank the Editor for the comment. We have added a new section to the Methods, defining the fractional difference between two arbitrary variables. This equation is then referenced at various relevant points throughout the text and in the Supplementary Material.